# Situation Awareness-Oriented Patient Monitoring with Visual Patient Technology: A Qualitative Review of the Primary Research

**DOI:** 10.3390/s20072112

**Published:** 2020-04-09

**Authors:** David Werner Tscholl, Julian Rössler, Sadiq Said, Alexander Kaserer, Donat Rudolf Spahn, Christoph Beat Nöthiger

**Affiliations:** Institute of Anesthesiology, University and University Hospital Zurich, Rämistrasse 100, 8091 Zurich, Switzerland; julian.roessler@usz.ch (J.R.); sadiq.said@usz.ch (S.S.); Alexander.kaserer@usz.ch (A.K.); donat.spahn@usz.ch (D.R.S.); christoph.noethiger@usz.ch (C.B.N.)

**Keywords:** Visual Patient, monitoring, avatar-based technology, situation awareness

## Abstract

Visual Patient technology is a situation awareness-oriented visualization technology that translates numerical and waveform patient monitoring data into a new user-centered visual language. Vital sign values are converted into colors, shapes, and rhythmic movements—a language humans can easily perceive and interpret—on a patient avatar model in real time. In this review, we summarize the current state of the research on the Visual Patient, including the technology, its history, and its scientific context. We also provide a summary of our primary research and a brief overview of research work on similar user-centered visualizations in medicine. In several computer-based studies under various experimental conditions, Visual Patient transferred more information per unit time, increased perceived diagnostic certainty, and lowered perceived workload. Eye tracking showed the technology worked because of the way it synthesizes and transforms vital sign information into new and logical forms corresponding to the real phenomena. The technology could be particularly useful for improving situation awareness in settings with high cognitive demand or when users must make quick decisions. This comprehensive review of Visual Patient research is the foundation for an evaluation of the technology in clinical applications, starting with a high-fidelity simulation study in early 2020.

## 1. Introduction

Visual Patient technology is a situation awareness-oriented visualization technology for patient monitoring developed by our research group at the Institute of Anesthesiology, University and University Hospital Zurich, Switzerland. The technology translates numerical and waveform patient monitoring data into a new user-centered visual language. Vital sign values are converted into colors, shapes, and rhythmic movements—a language humans can easily perceive and interpret—on a patient avatar model in real time. In several computer-based studies, anesthesia providers were able to perceive more monitoring information with lowered perceived workload and with increased perceived diagnostic certainty with Visual Patient, compared to conventional number- and waveform-based patient monitoring. In early 2020, we will begin a high-fidelity simulation study with a product by Philips (Koninklijke Philips N.V., Amsterdam, the Netherlands) developed from the Visual Patient concept. With this milestone, we move from basic research on our product [1,2,3,4,5,6,7] to clinical application research. This paper contributes a comprehensive overview of the research behind Visual Patient to the current literature. In the introduction (Section 1.1, Section 1.2, Section 1.3, Section 1.4 and Section 1.5), we summarize the scientific context of the technology, situation awareness, flight visualization technology, patient monitoring, and explain how Visual Patient rests in it. Section 2 deals with the characteristics and limitations of Visual Patient. In Section 3, we summarize the previous research work qualitatively. Section 4 synthesizes the findings of our work and discusses them in the view of measured outcomes, essential issues in patient monitoring, and other similar visualization technologies. Visual Patient is a user-centered, situation awareness-oriented technology; therefore, the methods and findings described in this paper are relevant in general in the development of indicators that rely on efficient information transfer. When humans are involved in decision making, a sensor can only be as robust as the indicator that displays the information obtained. Therefore, we believe this paper is placed accurately in Sensors.

### 1.1. Situation Awareness

The concept of situation awareness, also known as situational awareness, is at the heart of the Visual Patient technology. The concept originates from aviation, but it is the underlying driver of successful decision making in many different fields, including the health care sector [8,9,10,11].

The three levels of situation awareness are “the perception of elements of the environment within a volume of time and space, the comprehension of their meaning, and the projection of their status into the near future” [8,9]. The first level of situation awareness is the perception of environmental elements; for example, in the setting of anesthesia that corresponds to noticing the status and dynamics of the patient’s vital signs on the monitor. Research has found up to 80% of anesthesia adverse events result from a lack of situation awareness, with level 1 errors (perception) accounting for the largest share (42%) [12,13]. The second and third levels of situation awareness are the comprehension of these perceived elements and their projected course. The correct understanding of a situation (a care provider’s mental model of that situation) forms the basis of sound clinical decision making, clinical performance, and, ultimately, patient safety [8].

### 1.2. State-of-the-Art Patient Monitoring

#### 1.2.1. Scope of the Area of Application

A patient monitor measures and displays a patient’s vital signs, which are measured by different sensors. The monitor enables the caregiver to take corrective actions quicker than would be possible through observation and assessment of clinical signs alone. Noninvasive monitoring offers an excellent risk–benefit ratio because it is virtually risk-free, yet it can prevent potentially catastrophic complications such as brain damage [14,15,16,17,18]. In 2012, the last year that high-quality data are available, there were approximately 313 million surgical procedures performed worldwide [18]. The World Health Organization considers complications in the perioperative setting a major cause of avoidable death and disability around the world. The World Health Organization’s guidelines for safe surgery consider continuous patient monitoring by a trained and vigilant care provider during surgical procedures “extremely important” for patient safety [14]. Patient monitoring is equally useful in intensive care and emergency medicine. With continuous technological advances in sensor and microcomputer technology, patient monitoring will likely expand into additional areas where patients are not currently continuously monitored, such as regular hospital wards and retirement homes [19,20,21,22,23].

#### 1.2.2. Limitations of Current Patient Monitors

Patient monitoring is the interface between the physical quantities measured in the patient on one side and the sensorium and cognition of the human decision makers on the other. However, the interface design of current patient monitors neglects the strengths of human sensory perception and does not promote an optimal understanding of the patient’s condition in a time-efficient manner [24]. The visual system of humans is better suited for recognizing shapes, colors, and movement than for reading numbers. With traditional monitors, care providers must invest high cognitive effort to integrate the presented information into a mental model of the patient’s current status and expected progression. Several characteristics of the conventional number- and waveform-based representations of vital signs are responsible for these shortcomings. In fact, the presentation of patient-monitoring information has not evolved since the first electrocardiograms in the first half of the 20th century. In a single-sensor-single-indicator fashion, the various data streams measured by sensors are displayed as individual numbers and waveforms. An exception to this is the display of trend information, which shows the course of the measured values as graphic bands. The single-sensor-single-indicator arrangement is a rudimentary, technology-centered way of information presentation. It corresponds to an old-school conventional cockpit where many individual instruments indicate measurements from individual sensors. These interface designs cause difficulties in the reception of information by human users. First, we can only read numbers using foveal or sharp vision, which we can only direct at one digit at a time [25]. Second, to get a complete picture of the patient’s situation, we must first read and cognitively integrate all the numbers displayed on the monitor before we can derive meaning. Third, many of the displayed vital signs can have the same value; for example, pulse rate, electrocardiogram heart rate, blood pressure, oxygen saturation, neuromuscular relaxation, and brain activity may all have a value of 94 (or 94%). Finally, people can only remember seven digits plus or minus two in their short-term memory [26,27]. These human performance limitations force care providers to resort to piecewise data gathering and assembling of these pieces to build a mental model of the situation (situation awareness).

#### 1.2.3. Patient Monitoring in Clinical Reality

Patient monitoring contributes significantly to the safety of modern medicine. Care providers base their therapeutic plans on data obtained from patient monitoring. However, real-world studies have identified problems with patient monitoring due to at-a-glance monitoring, inattentional blindness, and alarm fatigue. Studies by Ford et al. [28] and Loeb [29] on the monitoring behavior of anesthesiologists found that care providers observe the patient monitors in glances of one- to two-second durations (“at-a-glance” monitoring). Other research has found that increasing the amount of information displayed on a monitor reduces the ability of users to detect unexpected changes, even when they are in plain view (inattentional blindness) [30,31,32,33]. Frequent acoustic and visual alarms from patient monitoring lead to alarm fatigue. Correct alerts are no longer perceived because the practitioner becomes dulled [34,35,36,37,38,39,40]. Distractions, such as background music, alarm sounds, and other cognitive and emotional stimuli, are ubiquitous in operating rooms and intensive care units and cause attention consumption, which reduces the cognitive capacity of care providers for the complex task of patient monitoring [41]. Based on the feedback of 137 anesthesiologists and certified anaesthesia nurses in a mixed qualitative and quantitative study by our research group [42], we defined the following qualities of an ideal monitor: one that does not create false alarms, operates without disturbing cables, and transmits information quickly and easily. These qualities would help sustain user alertness and performance.

### 1.3. Visual Patient Technology

Visual Patient technology is a situation awareness-oriented visualization technology for patient monitoring. It translates numerical and waveform data into a new user-centered visual language that we developed based on aviation’s most visible situation awareness-oriented technology—synthetic vision—and in accordance with research results in other fields of science. The technology’s primary intended purpose is to communicate the patient’s condition to caregivers as quickly as possible and with as little cognitive effort as possible. This aim follows the goal of user-centered design as defined by Mica Endsley and Deborah Jones: “Our goal is to create system interface designs that transmit needed information to the operator as quickly as possible and without undue cognitive effort” [8]. 

### 1.4. Synthetic Vision, Aviation, and Visual Patient

Synthetic vision is a flight visualization technology pioneered by NASA (National Aeronautics and Space Administration) and the United States Airforce in the 1970 s and 1980 s [43,44]. Due to limitations in microprocessor computing power and display technology, synthetic vision only became commercially available in the mid-2000 s. Synthetic vision generates a virtual image of the flight situation from Global Positioning System geolocation data, terrain and airport data, aircraft attitude (orientation in space), traffic data, and other data available in an aircraft. For pilots, the resulting virtual image looks as if they were looking out the windshield on a clear day. A lake looks like a lake, a mountain looks like a mountain, traffic is displayed where it is, and so on. With synthetic vision, the flight situation can be interpreted more intuitively, more quickly, and with greater confidence than when pilots had to cognitively assemble this “picture” piecemeal from lower-level data from conventional instruments [45].

In the 15 years since its commercial release, synthetic vision has taken the aerospace industry by storm. The technology is routinely used: from iPad-based mobile flight applications (e.g., Foreflight by Boeing Corp., Chicago, IL, USA) through the smallest microlight aircraft, the entire general and business aviation (e.g., Bombardier Inc., Montreal, QC, Canada, Global 7500) segment to the largest long-range passenger aircraft (e.g., Airbus S.A.S. Toulouse, France, A350) and helicopters (e.g., Airbus H145).

The idea of Visual Patient technology was born from aviation and synthetic vision. Upon seeing synthetic vision technology in a Garmin G1000 integrated avionics system for the first time in 2012, author D.W. Tscholl wondered why we did not have this for our patients. When presented with the Visual Patient concept, Immanuel Barshi of the Human System Integration Division at NASA Ames (Moffet Field, CA, USA) said, “You got to try it,” and development began. Visual Patient technology uses the same logic as synthetic vision by creating a virtual image of the patient from vital sign patient monitoring data. It presents the data in a way that corresponds to the real phenomena as they are being measured—just like synthetic vision. The technology translates vital signs onto a patient avatar model in real time from the incoming stream of monitoring data. In short, Visual Patient technology is a research-driven development that applies scientific findings to improve the perception of patient monitoring information by human users.

### 1.5. Design Philosophy and Scientific Context

We drafted the initial version of the avatar and its visualizations based on principles of logic, user-centered design, and human–computer interaction, specifically, the picture theory of representation from the Tractatus Logico-Philosophicus by Wittgenstein [45], principles of user-centered design by Endsley [8], and results from the NASA publication “On Organization of Information: Approach and Early Work” by Degani et al. [46]. According to Wittgenstein’s theory, a picture is logical when it depicts a model of reality (i.e., it has a meaningful commonality with the reality it attempts to reflect) [45]. Principles of user-centered design recommend the use of direct representations of the phenomena that are causing the information to facilitate the creation of situation awareness [8]. The NASA publication includes outlines of the different hierarchical levels of representation and presentation of information, with the highest level of “order and wholeness,” achieved by integrating needed information in a single display [46]. From this work, we developed Visual Patient avatar version 1, which we used in the first round of the iterative development process, summarized subsequently.

## 2. Characteristics of the Visual Patient Technology

The following five characteristics of the Visual Patient technology result from previous research by other researchers and results of our research: (1) display of high-order information synthesized in one picture, (2) preprocessing (simplification) of data, (3) direct (logical) presentation of information, (4) parallel information transfer, and (5) peripheral vision monitoring. These characteristics lead to advantages and some context-specific disadvantages of the technology. The avatar version used in all our studies to date displays the 11 most frequently monitored vital signs, which are outlined and described in Table 1. A graphical example of the avatar is provided in Figure 1, and an audiovisual explanation is available in Appendix A. The avatar depicted in Figure 1 indicates low oxygen saturation (cyanotic skin) and body temperature (ice signs), high blood pressure (body expansion), expiratory carbon dioxide (gas cloud), and brain activity (eyes open). Furthermore, it indicates ST-segment abnormality (heart muscle hypoxia indicator) and regular tidal volume (lungs extend to the white line), central venous pressure, and neuromuscular function. The pulse rate and respiratory rate require an animation for interpretation. The pulse rate and respiratory rate require an animated sequence for interpretation.

### 2.1. Display of High-Order Information Synthesized in One Picture (Multiple-Sensor-Multiple-Indicator Philosophy)

The Visual Patient technology displays multiple vital signs integrated into a single indicator (the patient avatar). For example, the caregiver can evaluate the respiratory rate based on the respiratory rate of the avatar’s lungs and the rate of formation of the exhaled carbon dioxide (CO_2_) cloud. With a single look at the avatar’s body, a caregiver can interpret the pulse rate (pulsation frequency of the body), oxygen saturation (skin color of the avatar), blood pressure (pulsation intensity of the body), neuromuscular relaxation (form of the body), and body temperature (heatwaves or ice crystals present or not). This eliminates the piecemeal process of information gathering in conventional patient monitoring, in which multiple numerical values have to be read individually one after the other before they can be decoded for meaning.

### 2.2. Preprocessing (Simplification) of Data

The avatar technology preprocesses the data for each vital sign into the following categories: no data measured, too low, normal, or too high. The technology reduces the complexity of the vital sign data by translating multiple continuous numerical values into two or three discrete visualization conditions. With all vital signs, the avatar can display a total of 30 different visualization conditions, thereby rendering the technology capable of displaying 4608 different situations. This number excludes cases in which one or more sensors do not provide any data. Information preprocessing leads to improved comprehensibility and diagnostic certainty but reduces data precision (three discrete categories versus hundreds of separate numbers in the case of blood pressure values) [45]. Therefore, users may need to quantify the extent of a vital sign deviation using the numerical indicator. We understand the avatar as a complementary technology to routine patient monitoring. It intends to help care providers to more efficiently perceive critical information that is already available in the patient monitoring system.

### 2.3. Direct (Logical) Presentation of Information

Direct presentation of information eliminates the need for service providers to calculate the required information (e.g., What is the current depth of anesthesia?) from low-level data (e.g., bispectral index of 75) [46]. The same information on the avatar is displayed as eyes open, corresponding to the mental model of a patient that is awake. On the avatar, the individual vital signs correspond to elements whose representation reflects the expected real phenomena associated with a particular patient status. We modeled the visualizations according to the mental models of the care providers. For example, the avatar’s body can pulse slowly, regularly, or quickly according to the measured pulse rate, which represents the pulse wave that passes through the real patient’s body. Also, the avatar may have a natural or purple skin color, representing normal oxygen saturation or hypoxemia, or the avatar may exhale a small, regular, or large cloud of CO_2_, depending on the measured exhaled CO_2_ concentration.

### 2.4. Parallel Information Transfer

Because of the use of colors, shapes, and movements, users can perceive several vital signs simultaneously, according to the results of one of our eye-tracking studies [3].

### 2.5. Peripheral Vision Monitoring

We generated empirical evidence for peripheral vision monitoring through eye-tracking studies [4]. Because of the way the avatar displays the information as colorful moving graphical objects, caregivers can monitor some of the patient’s vital signs using only their peripheral field of view. Conventional patient monitoring is particularly unsuitable for patient monitoring with peripheral vision because its interpretation involves the reading of a large quantity of individual numbers. To read numbers from a patient monitor, care providers must fix their foveal or sharp vision directly on the number they intend to read. Outside the small central area of the foveal view, visual acuity deteriorates so glyphs cannot be read [25,27].

### 2.6. Limitations of Visual Patient

An inherent limitation of the avatar design is the pre-processing of vital signs into categories. Visual Patient removes the process of simplifying the vital signs into the categories “too low”, “normal”, or “too high” from the users. This pre-processing leads to improved comprehensibility and diagnostic certainty but, at the same time, reduces data accuracy. For example, the pulse rate in the current avatar version can only assume one of three individual states. Conversely, monitoring with numbers can indicate about 300 different values between 0 and 300. After detecting a deviation with Visual Patient, it may be necessary to follow it up in its exact numerical form.

Another limitation of the Visual Patient version used in the reviewed studies is that it cannot yet display trends. This aspect is essential because trend displays of conventional patient monitors can help healthcare providers to detect slow changes over time, which often serve as early warning signals before vital signs reach a too high or too low alarm. Therefore, the avatar cannot replace routine monitoring but may serve as a supplement that explicitly aims to improve situation awareness. The optimal integration of the two technologies will be the key to success, just as with synthetic vision and numerical flight data. 

## 3. Qualitative Review of the Studies Conducted to Date

We divided the Visual Patient studies published so far into two major series: Visual Patient series 1 and Visual Patient series 2. Series 1 contains the preliminary studies, which aimed at qualitative and quantitative validation, stepwise improvement, and the first comparison of the technology’s performance with regular monitoring [1,2,3]. In series 2, we tested the technology in three particular conditions for patient monitoring: using only peripheral vision [4], while being distracted [6], and when monitoring multiple patients at the same time, such as is in an intensive care unit central station monitor [5]. Both study series included computer-based evaluations of monitoring scenarios by the study participants and assessment of eye-tracking data and mixed qualitative and quantitative analyses of user impressions. We conducted all studies in series 1 and 2 as two-center studies in Switzerland at the University Hospital of Zurich and the Cantonal Hospital of Winterthur. Additionally, we review a study in which we compared two different methods of training in the technology: individual personal training and training a class in an auditorium [7]. A summary of all studies is provided in Table 2. 

### 3.1. Data Collection Tool

For the experiments described in this literature review, we used an iPad- (Apple Inc, Cupertino, CA, USA) and iSurvey- (Harvestyourdata.org, Wellington, New Zealand) based data collection tool, which we developed and validated in previous clinical studies [47,48,49].

### 3.2. Visual Patient Series 1

#### 3.2.1. Development and Validation of the Visual Patient Design

The first study was a validation study by Tscholl et al. published in 2018 [1]. This study aimed to improve the initial version of the Visual Patient avatar in successive steps until all visualizations reached interrater reliability of more than 94%. The initial design was drafted according to the principles of situation awareness-based design and with the involvement of various human factors experts (psychologists, physicians, pilots, and software developers). The study tested three successive versions of the Visual Patient avatar with 150 anesthesia providers. To measure interrater reliability, the participants evaluated several monitoring scenarios featuring randomly combined avatar-based visualizations after watching a training video (Appendix A) and jointly evaluating one monitoring scenario with the data collector. In the third and final version tested, each of the avatar’s 30 visualizations reached interrater reliability of more than 94%, which yielded a global Fleiss’ kappa of 0.98 (95% CI [confidence interval] 0.96–0.99, *p* < 0.001). This paper included a qualitative analysis of the participant feedback. This analysis led to three significant design modifications. 1. We changed the body color for low oxygen saturation from an intense blue to purple. Several participants stated that the original vivid blue color made them assume a hypothermic (cold) patient. 2. We added a thin, always visible, white line, which indicates the safe state for all vital signs visualized by the expansion of an element of the avatar. We have added this modification to prevent possible interactions of visualizations with each other and to provide users with an apparent reference for the normal state. 3. We implemented a shadow, which designates the maximum of the current extension of an element. We added this modification for all vital signs, which are visualized through the extension of an element to enable instant recognition, at any point during the pulsation- or breathing cycle. This modification resulted from feedback indicating that a slow pulse and respiratory rate would increase the time to recognize blood pressure, tidal volume, and expiratory CO_2_.

The International Medical Informatics Association included this paper in its 2019 Yearbook of Medical Informatics as one of 2018′s best human factors papers.

#### 3.2.2. Visual Patient Versus Conventional Monitor

This first comparative study was part of the first validation study [1]. Thirty-two anesthetists participated in this computer-based study with a within-participant study design. The participants included 12 senior physicians, eight resident physicians, and 12 certified anesthesia nurses. They received the same scenarios alternating between avatar-based and regular presentation. To test the technology’s performance in transmitting essential monitoring information, the study participants evaluated short patient monitoring scenario videos of 3 and 10 s duration. After each video, the participants recalled the status of the 11 vital signs shown and assessed how confident they were that they remembered each vital sign correctly. Once per scenario, the participants evaluated the subjectively perceived workload using the NASA Task Load Index (TLX) questionnaire [50,51,52]. This study found that the number of vital signs perceived with the avatar in comparison with conventional patient monitoring more than doubled in the 3 s scenarios and just slightly less than doubled with the 10 s viewing durations. With the avatar, the median confidence ratings were “confident” or higher in all situations, whereas median confidence was “unconfident” in all 3 s scenarios with regular monitoring. Participants perceived more vital signs in 3 s of avatar-based monitoring and, at the same time, rated experienced workload lower than in 10 s of routine patient monitoring. Participants’ confidence ratings improved with avatar-based monitoring compared with conventional patient monitoring in all scenarios. In this study, *p* values were ≤ 0.006 for all paired Student’s *t*-tests. 

#### 3.2.3. User Perceptions

After the data collection sessions for the validation study described previously, we conducted semistructured interviews with most of the study participants (128 of 150, 85%) [2]. The data collector asked the participants to openly express their opinions about the two interview questions: “Which advantages do you see in Visual Patient monitoring technology and why?” and “What should we improve in Visual Patient?” Using qualitative analysis, we derived higher-level themes and subthemes from the participants’ responses. The results to the “Which advantages do you see in Visual Patient monitoring technology and why?” question were published in 2018 [2]. The results to the “What should we improve in Visual Patient?” question were published in the Appendix A of the first article [1]. In a second step, we defined one statement, which we considered crucial for the better understanding and further development of the technology, for each of the four identified high-level topics: (1) quick situation recognition, (2) intuitiveness, (3) unique design characteristics, and (4) potential future uses. A new group of 36 anesthetists (79% in this group had not participated in the interviews) who participated in Visual Patient series 2 studies, then rated their agreement with these statements on Likert scales administered in the form of a poststudy online survey. In this survey, 82% percent of participants agreed to the statement “I found Visual Patient technology to be intuitive and easy to learn,” and 63% agreed with the statement “Visual Patient technology enabled me to get a quick overview of the situation.” Fifty-three percent agreed to the statement “I think Visual Patient technology might be helpful for nonexperts in patient monitoring in the health care system.” Only 11% agreed to the statement “The visual design features of Visual Patient technology are not helpful for patient monitoring.” Through the quantification of the participants’ agreement or disagreement with these statements, this study achieved a higher level of evidence than the purely qualitative description alone would have. All Wilcoxon signed-rank tests performed to analyze whether the medians of the responses differed significantly from the neutral options gave *p* values of < 0.01. 

#### 3.2.4. Eye Tracking

In this study published in 2020 [3], we analyzed the eye-tracking data collected during the Visual Patient comparative study described previously [1]. Using a stationary eye tracker, we recorded the study participants’ gaze fixation data and eye movements while they watched the monitoring scenarios. We analyzed which vital signs the study participants fixated on and for how long in the regular and avatar-based monitoring scenarios. To validate the methodology, we analyzed the correlations between visually fixated and correctly recalled vital signs. This experiment considered neurophysiological principles teaching that glyphs (a letter or a number) can only be read and thus potentially understood if observed with foveal vision. Foveal vision is the tiny area in the center of the visual field in which we see sharply enough to be able to read. At an arm’s length away from a monitor, foveal vision corresponds to an area of approximately two centimeters in diameter. Multivariable linear regression revealed that the type of monitoring technology (routine number- and waveform-based versus avatar-based) was an independent predictor of the number of visually fixed vital signs (more with avatar-based monitoring). The difference was more substantial in the shorter 3 s scenarios than in the extended 10 s observations. The study center, profession, gender, and the order with which the videos appeared did not affect the results. In every scenario, the participants observed nine of 11 total vital signs statistically significantly longer with the avatar. The critical vitals (pulse rate, blood pressure, oxygen saturation, and respiratory rate) remained visible for almost the entire scenario duration with avatar-based monitoring but only for a fraction of the time of a scenario with regular monitoring. We found that the visual fixation of a vital sign correlated in both technologies with the correct recall of this particular vital sign. 

With this eye-tracking study, we added a layer of evidence applying a new quantitative method to explain one of the mechanisms by which avatar-based monitoring improves the perception of vital signs. In regular patient monitoring, participants read the information number by number, one after the other. However, in avatar-based patient monitoring, information about several vital signs can be read with every single visual fixation. The technology transmits vital sign information in parallel because it transforms it from numerical and waveform format into forms, colors, and frequencies. For example, a single glance at the avatar’s body provides information about pulse rate (frequency of pulsation), blood pressure (intensity of pulsation), oxygen saturation (skin color), neuromuscular relaxation (floppy or stiff extremities), and body temperature (heatwaves or ice crystals).

### 3.3. Visual Patient Series 2

#### 3.3.1. Peripheral Vision

This eye-tracking study by Pfarr et al. published in 2019 [4] follows the same theoretical background as the Visual Patient series 1 eye-tracking study [3]. According to these neurophysiological principles, a person can only read glyphs when they look at them with foveal or sharp vision. The foveal field of view corresponds to a small central part of the visual field with the approximate size of a thumbnail at an arm’s length from the eyes [27]. In this experiment, the participants sat in front of a computer screen at a distance of approximately one arm’s length (60 cm) and looked at the image of an animated cat. To ensure that the participants maintained their foveal view of the animated cat, we recorded their eye movements with a stationary eye tracker. At an angle of 45 degrees, this screen stayed in the peripheral field of view (30–60 degrees lateral) of the participants. On the monitor to the left, we played the patient monitoring scenarios, alternatingly between number- and curve form-based or avatar-based. After 5 s, one of the 11 displayed vital signs turned abnormal, and we asked the participants whether they had recognized the parameter that changed and the direction in which it moved. Furthermore, the participants rated how confident they were in their answers. Of 30 participants, 28 achieved an improved result with avatar-based patient monitoring. In addition, participants’ perceived confidence in the correctness of their diagnoses was higher for 29 of 30 participants for avatar-based monitoring. Only one participant rated their confidence higher for conventional monitoring. As previous studies found, anesthesiologists only look directly at their patient monitors for short glances during anesthesia cases [28]. These findings mean that anesthesia providers spend much of their time at an angle where patient monitoring with peripheral vision would, in theory, be possible. Our results suggest that peripheral vision vital sign monitoring could be suitable for patient monitoring when using avatar-based displays [4].

#### 3.3.2. Distractions

In the operating room, the care providers face various distractions, including acoustic, visual, cognitive, and emotional stressors. Distractions are a safety-relevant factor because they may impair vigilance, situation awareness, and, most importantly, decision making [34,35,36,37,38,53]. Therefore, we wanted to test whether avatar-based monitoring could improve the perception of vital signs and situation awareness under distraction compared with regular monitoring. The hypothesis arose because avatar-based monitoring does not work by memorizing numbers but by interpreting colored objects, which we suspected to be less prone to distractions. To test this hypothesis, we conducted the following experiment, the results of which were published in 2019 [6]. The participants sat with a data collector in a quiet room where they could watch and evaluate monitoring scenarios undisturbed. We played various scenarios, alternating between avatar-based and conventional monitoring. All situations showed 11 vital signs, some normal and some pathological. After 3 or 10 s, the screen turned black, and the participants had to recall as many of the 11 vital signs as possible, choosing between “too low,” “safe,” “too high,” or “no recall.” The Paced Auditory Serial Additions Test (PASAT), a simple arithmetic task, served as the standardized distraction and was performed by the participants during half of the scenarios [54]. The other half of the situations did not include a distraction. The participants watched the same scenarios with and without distraction in regular and avatar-based monitoring. After each scene, the participants completed the NASA TLX questionnaire. Thirty-eight participants took part in the study. In all cases, participants remembered more vital signs with avatar-based monitoring both with and without distraction compared with conventional patient monitoring. The standardized disturbance reduced performance and increased perceived workload in all situations compared with the undisturbed version. These results revealed the importance of avoiding unnecessary distractions in the operating room as much as possible, while providing care providers with the simplest-to-interpret monitoring possible. The benefits of avatar-based patient monitoring may be especially helpful in high-workload situations when cognitive resources and task performance are reduced [40,55,56,57].

#### 3.3.3. Monitoring Multiple Patients

On central station patient monitors, doctors and nurses in intensive care units and operating rooms see dozens of vital signs for multiple patients displayed on a single, large screen. We hypothesized that avatar-based monitoring might be especially useful for easy and intuitive information transfer in order not to overlook important information. To test this hypothesis, we conducted a within-subject, computer-based laboratory study, which was published in 2020 [5]. We showed each participant four different central monitor scenarios in sequence, each situation displaying two critical and four healthy patients simultaneously for either 10 or 30 s. After this time, the screen turned black, and the participants recalled the vital signs of the two critical patients. We measured the perceived workload with the NASA TLX questionnaire. In the 10 s scenarios, the median number of perceived vital signs significantly improved from 7 to 11 using avatar-based versus regular monitoring. However, the perceived workload was 10% lower with avatar-based patient monitoring. In the 30 s scenarios, vital sign perception and the workload with avatar-based patient monitoring did not differ significantly. This study provides empirical evidence that avatar-based patient monitoring may help to improve situation awareness and reduce the workload when monitoring multiple patients at the same time, especially when observing the monitor for short durations.

#### 3.3.4. Individual Versus Class Instruction in an Auditorium

This study was published in 2020 [7]. Although previous studies have shown that Visual Patient is intuitive to understand, we aimed to compare different instruction methods for the novel technology [1,2,58]. We compared two groups, one of which received one-on-one instruction on Visual Patient, and the other underwent large-scale classroom instruction. As Visual Patient can be implemented in institutions with larger groups of personnel, it must be teachable to multiple people at once. We recruited 42 anesthesia professionals to the classroom group, who underwent 30 min of classroom-based instruction on Visual Patient. The comparison group was a historical sample from a previous Visual Patient study [1], in which the participants received individual instruction. We then used two-way mixed ANOVAs and mixed models to compare the ability to correctly interpret vital signs in a simulated monitoring scenario in both groups. There was a statistically significant interaction between the teaching intervention and display technology on perceived vital signs. The mixed logistic regression model for correct vital sign perception yielded an OR of 1.88 (95% CI 1.41–2.52, *p* < 0.0001) for individual instruction compared with class instruction. We found an OR of 3.03 (95% CI 2.50–3.70, *p* < 0.0001) for correct vital sign perception with Visual Patient compared with conventional monitoring. These results illustrate that although individual instruction is slightly better, classroom-based instruction is a viable alternative and produces satisfactory results in the adoption of Visual Patient. This makes Visual Patient suitable for large-scale implementation in health care organizations. Further research may evaluate other methods of instruction, such as e-learning, or if instruction is necessary at all (commonly used monitoring equipment is often not formally introduced to novel users).

## 4. Discussion

This review provides a condensed summary of the background and current state of the research regarding Visual Patient technology, including its defining properties and characteristics. In our primary research so far, we measured the effects of Visual Patient technology on outcomes closely related to the concept of situation awareness: perception of vital signs, perceived diagnostic certainty, perceived workload, and the intuitiveness and learnability of the technology [8,9,10,24,59,60]. The following is a discussion of the performance of the technology and our experiences during data collection for the studies.

### 4.1. Vital Sign Perception

In five separate experiments, we compared the perceptual performance of the participants with Visual Patient and routine patient monitoring [1,4,5,6,7]. The participants looked at scenarios in randomized order and had to indicate how they perceived the vital signs after the short observation periods. The tested scenes were visible for between 3 and 30 s and contained distractions and central monitor scenarios featuring multiple patients. In the 3 s scenarios, we found the greatest differences between technologies: participants were able to perceive more than twice as many vital signs with Visual Patient [1,3,6]. In the 10 s scenarios, with and without distraction and in single and multipatient situations, we also found clinically and statistically significant improvements [1,4,5,7]. The only situations where there was no statistically significant difference were the 10 s classroom instruction scenario and the 30 s central monitor scenarios with several patients [5]. We validated these findings using eye tracking and mixed qualitative and quantitative analyses [2,3,4].

In an eye-tracking study [3], we found that the vital signs that the participants fixed on visually were also those that they assessed correctly. The data showed that with avatar-based patient monitoring, participants were able to perceive information about several vital signs with every single visual fixation. In contrast, with conventional glyph-based patient monitoring, they were only able to read the vital sign sequentially, one per visual fixation. We gave this underlying mechanism of the Visual Patient technology the name “parallel information transfer”. This mechanism is possible because the avatar technology transforms the information from numerical and waveform format into forms, colors, and frequencies, which participants can interpret in parallel. [3].

In a second eye-tracking study, we showed the peripheral field of vision becomes available for patient monitoring with Visual Patient [4]. This characteristic suggests exciting theoretical possibilities for patient monitoring of the future; for example, the patient will always be in view when users look at the monitor with foveal view and with their peripheral vision. Several studies reported that anesthesia providers tend to look at patient monitors for brief moments of time during an operation [28,29]. For most of the operation time, the patient monitor is in the care providers’ peripheral field of view.

Peripheral vision could have aided perception with Visual Patient in our studies [1,2,3,4,5,6,7]. When users look at a point on the monitor with foveal vision, information is still transferred from outside the foveal area with Visual Patient, such as the color or pulsation frequency of the body. These perception improvements are reflected in the mixed quantitative and qualitative analysis of the study participants’ interview responses, in which two-thirds of the participants agreed with the statement “The technology enabled me to get a quick overview of the situation” [2].

We measured information perception (first level of situation awareness) because most human errors are due to situation awareness level 1 errors, in which available information is unrecognized [8,12,13]. This fact underscores the dire need for information tools that are simpler to interpret. Cognitive aids, such as checklists, improve situation awareness and have been found to improve performance in a variety of medical applications [48,61,62,63]. Visual Patient aims to increase the perceptible amount of information and thereby enhance the mental model of the user, ultimately improving situation awareness on all levels.

### 4.2. Perceived Workload

To compare the perceived workload of the participants for both technologies, we asked them to complete the NASA Task Load Index questionnaire, a comprehensively validated tool for measuring subjectively perceived workload [50,51,52,64,65,66]. We used this tool in three of the seven studies. The rationale for measuring the perceived workload was that high cognitive workload reduces the capacity for information processing and, thus, generation of situation awareness, decision making, and task performance. Stress causes people to pay less attention to peripheral information, scan information in a more unorganized manner, and rush decisions without consideration of all available information. The participants perceived the workload as lower with the avatar in all 10 s situations. However, the perceived workload decreased most markedly—by more than 10%—in the 3 s scenarios.

### 4.3. Perceived Diagnostic Certainty

In two of the experiments, we measured the confidence of the participants in their diagnoses (perceptions). We measured this outcome because uncertainty is a psychological stress factor, and situation awareness and confidence work together to enable a person to make a decision. Research has shown that if situation awareness and confidence in that situation awareness are high, a person will more likely achieve a good result than if situation awareness is equally good but the person has less confidence in that situation awareness [8,67]. With low confidence, the person will more likely behave protectively and choose to gather more information and will be less likely to respond and thus be less effective. Visual Patient showed a higher perceived diagnostic certainty in the accuracy of the diagnoses. These results were also validated by an eye-tracking study [3], which found a significant correlation between a correct response and a visually fixed vital sign.

With Visual Patient, the median confidence rating was at least “confident” in all cases. With regular monitoring, this only occurred in the 10 s scenarios. In all experiments, the participants were more confident with Visual Patient than with regular patient monitoring, with the largest differences in the 3 s and peripheral vision situations [1,4,5,6].

### 4.4. Learnability

Judging by the results of the experiments, Visual Patient technology has shown good learnability. Participants were able to achieve better results with the technology, which most had never seen until immediately before the study, than with regular monitoring, which most had been using professionally for years. For the Visual Patient training, we used a 6 min instruction video, which explained all visualizations one after the other. After a single view, the study participants were already able to correctly recognize more vital signs in the 3 s scenarios after both individual and classroom instruction than with regular monitoring.

Intuitiveness is the property of a technology that makes it usable by users using mainly the unconscious processing of stored experiential knowledge. Cognitive ease in learning a new technology is important when introducing new technologies. An intuitive user interface creates confidence in a product and is vital for user acceptance. [59,60]. The statement with the highest agreement (82%) in our qualitative and quantitative user-perception study was “I found Visual Patient technology intuitive and easy to learn” [2].

### 4.5. Visual Patient and Relevant Patient Monitoring Aspects

In this section, we would like to discuss five crucial aspects of patient monitoring: alarm fatigue, artifacts, trends, pattern recognition, and event monitoring and explore how Visual Patient may interplay with these aspects.

#### 4.5.1. Alarm Fatigue

Alarm fatigue is an important topic that anesthetists, operating room, and intensive care unit staff consider problematic in patient monitoring [42]. When alarms sound too frequently, care providers may get numb and, as a result, may miss real alarms [68,69,70]. Also, disturbances from frequent alarms may put patients in intensive care units at increased risk for delirium. Studies found that in some cases 20% or less of all alerts had therapeutic consequences [69,70]. If Visual Patient is configured more conservatively than the audio alarms, it may help to reduce acoustic alarms because it can provide an earlier visual clue. Visual Patient this way may serve as an earlier line of defense before an audio signal goes off as a later defense level. Similar mechanisms of incremental alarms are implemented in airliner cockpits, too.

#### 4.5.2. Artifacts

Artifacts are another common problem in patient monitoring, which anesthetists find disturbing [42]. To understand how Visual Patient handles artifacts, we should consider several essential points. Visual Patient, at least in the version tested in the studies summarized in this review, is an indicator technology that corresponds more to a front-end than it does to a back-end. It is a technology that primarily aims to display information efficiently but not modify the incoming data streams. Accordingly, in the studies conducted thus far, we set up Visual Patient to correspond 1:1 to the classical alarm settings. With this setting, if an artifact would trigger an acoustic alarm, Visual Patient would have indicated it as well. Measurement artifacts may originate on the patient side (e.g., atrial fibrillation causing the heart rate values to move up and down through the set alarm threshold, or a sensor that fell off the patient) or the technology side (e.g., use of electrocautery with associated ECG disturbances). Users adjust the alarm limits accordingly or silence the generated alarms. While they can be annoying, they always happen for a reason, and we must display them as clearly as possible to the users if we aim to enable the best situation awareness and adequate response to an artifact (e.g., adjust the alarm threshold, reapply the sensor). Uncertainty is a psychological stress factor. In stressful situations, when the monitor rings several alarm bells, stress is caused by users’ feelings of uncertainty about what exactly is going on. Visual Patient may help to take some of the stress associated with artifacts off the care providers, because Visual Patient, renders the nature of the alarm more readily understandable.

#### 4.5.3. Trend Monitoring

The lack of trends is a limitation of the current Visual Patient version. Trends are vital signs graphically plotted along a time axis and often provide an early warning sign before a vital sign reaches a too high or too low alarm state. Our research group will begin to study a trend function for Visual Patient this year. Visual Patient Trend Preview, a promising algorithm for predicting the very near future from the very recent past. For the current version of Visual Patient, the same applies to the question of trend display as for the other limitations of the technology. Visual Patient does some things better than conventional monitoring but not everything. Optimal integration of both technologies will be the aim in going forward.

#### 4.5.4. Pattern Recognition

Pattern recognition in patients’ vital signs helps experienced clinicians to make decisions according to mental models they have stored for such situations [8,71]. We have not yet explicitly investigated the effects of Visual Patient on pattern recognition. However, from what we have learned about the technology, it seems that changes that are large enough that they will show up in Visual Patient might be detectable better in the direct visualization of Visual Patient than the abstract data values in regular monitoring. Subtle changes, on the other hand, might be detectable better in the higher data resolution of routine patient monitoring. In our simulations, we have noticed how Visual Patient may facilitate pattern recognition in certain situations. For example, in atrial fibrillation, a rapidly pulsating heart of the Visual Patient avatar, together with a more slowly pulsating body, illustrates the root problem very well (not every ECG heartbeat produces a pulse). The relationship between open eyes and relaxed body (risk of intraoperative awareness or residual neuromuscular relaxation in an awake patient) is much more impressive than reading a train-of-four ratio of 60% and a bispectral index of 73 from the regular monitor.

#### 4.5.5. Event Monitoring

Event monitoring is used in patient monitoring to display special events that monitoring software automatically detects. These include events as apnea and various arrhythmia events. Asystole and apnea already appear in Visual Patient. Visual Patient then does not pulsate or breathe. The user can distinguish this situation from the situation of a loose ECG cable because the heart and lungs are shown in peach color and are not grey and dashed. A peach color indicates that the wires are connected, but the heart is not beating / the patient is not breathing. But Visual Patient also seems to provide an ideal platform for the situation awareness-oriented presentation of other events. In a study that we currently prepare, we will test intensive care extensions for Visual Patient, including an upright and enlarged Visual Thorax. Visual Thorax includes anatomically correct ST-segment visualizations, electrical cardiac visualizations, and advanced hemodynamic visualizations. In the future, the possibilities of the Visual Patient Platform seem endless, just think of the information from the electronic patient data management system that is integrated into the visualization. We will finally be able to see at a glance where the installations of a patient are.

### 4.6. Impressions from Performing Studies with the Technology

The most notable impression we have is the feedback from the study participants. The technology seems to resonate with its human users. Although some participants appeared uncertain about what to expect at the beginning of the data collection sessions, not one participant was incapable of understanding the potential benefits of the technology. Therefore, over the years, participants have given Visual Patient many affectionate nicknames, including amoeba, voodoo puppet, jumping jack, patient mannequin, alien, ghost. That may be because the technology humanizes the patient, or because its design strikes a balance between a too realistic, anatomically correct patient and a too comic-like, abstract oversimplification.

Other points that came up frequently were why the patient seemingly lies on its head and why the head is so large. We wanted to represent as many different patients as possible, including babies and children with comparatively larger heads. The Visual Patient avatar is upside down because this corresponds to the common viewpoint for the anesthetist of the patient. The anesthetist usually stands on the side of the patient’s head. We chose the orientation of the Visual Patient to correspond with reality [8,45,46]. The orientation of the Visual Patient avatar will be changeable in a product, for example, for use in an intensive care unit, where doctors usually stand at the patient’s feet.

### 4.7. Strengths and Limitations of the Reviewed Studies

The studies were adequately powered from sample size planning based on pilot studies. All experiments also included intraparticipant comparisons, which reduced the potential effects of random noise. Except for one study that included only nurse anesthetists [7], all studies featured equal groups of professions (same number of senior physicians, residents, and nurses) and uniform gender distribution.

The gaps left unfilled by current Visual Patient research concern the transferability of the results of the primary research into clinical reality. Will it work as well in everyday clinical routine as in the studies? Further research should focus on the expansion and development of the technology. Can the technology incorporate meaningful trends, advanced vital signs, events, and other information like patient installations? Does it help with pattern recognition? A further focus could be on the application and learning of the technology. Can it be used by new user groups, such as paramedics, nurses without monitoring training, or completely untrained users? Does it work equally well in different cultures of the world? Although we assume humans can understand visual language more universally than numbers and letters, we still need to verify this. We will test the transferability of the results this year in our hospital with a realistic simulator study and a prototype patient monitor, and we will also begin a study this year evaluating additional visualizations for intensive care monitoring, as well as an algorithm based trend technology. We hope that when the product comes to the market, other research groups will conduct studies with this technology to help us develop it further and hopefully eventually test its impact on patient outcomes.

We plan to evaluate patient outcomes in a big data analysis of electronic anesthesia records before and after the introduction of Visual Patient in our hospital. This study would have the potential to show whether or not the technology improves real-life management.

### 4.8. Overview of Other Studies

This overview focuses on visualization technologies that use an avatar-based or anatomical approach to display patient monitoring data. For this area, which we have been researching more than seven years and in which we know the scientific as well as the patent literature, it is comprehensive. It only touches the field of advanced displays in the medical field, as this was not the focus of this paper. Here we would like to refer to an excellent systematic review by Drews and Westenskow [24].

Since the early 1990s, research groups around the world have been developing innovative technologies that aggregate several measurements into one picture to transform number- and waveform-based data into formats that can be more easily understood by care providers [72,73,74,75,76,77,78,79,80,81]. These studies, as well as our Visual Patient and Visual Clot projects [82,83], have shown advantages of alternative presentations for accuracy of diagnosis, time to diagnosis, and perceived workload. Visual Clot is a situation awareness-oriented visualization for rotational thromboelastometry outputs. This visualization technology creates a visual representation in the form of an animated 3-dimensional blood clot. In a prospective international dual-center study with 60 doctors, the overall proportion of correct therapeutic decisions was 100%, IQR (interquartile range) 83–100%, for Visual Clot compared to 44%, IQR 25–50%, for the standard presentation of the results, *p* < 0.001. Mixed regression models showed the decision time was 18.7 s shorter (95% CI 16.4−21.1), *p* < 0.001. The perceived workload was reduced, and the participants rated their diagnostic confidence with Visual Clot as higher, both *p* < 0.001. In the 2000 s and 2010 s, surprisingly few researchers have tested user centered interface designs in the medical field. One notable exception is the research group around Wachter and Westenskow [75,81], who created a display that anatomically represents lung function data and concluded that the graphical pulmonary display may serve as a useful adjunct to traditional displays in identifying adverse respiratory events. A key finding of their studies, which was also instrumental in the development of our technology, was that the presentation of information according to the human anatomy appears to be particularly well-suited for intuitive understanding by users [75]. Another group tested the effects of a multiparameter, intraoperative decision support system with real-time visualization [76]. The technology displays information on the organ systems of a schematized patient model, assigning the information to the appropriate organ systems. This group found that the use of this decision support system was associated with improved process measures, but not postoperative clinical outcomes. Only now are avatar-based technologies finding their way into daily clinical practice. Examples are the Dynamic Lung product from Hamilton (Hamilton Medical AG, Bonaduz, Switzerland), the AlertWatch product from Capsule (Capsule Technologies, Corporation, Aldover, MA, USA), the Visual Alarm product from Masimo (Masimo Corporation, Irvine, CA, USA), and the Visual Decision Support product from Edwards Lifesciences (Edwards Lifesciences Corporation, Irvine, CA, USA). More research is needed to investigate the value of these technologies in the real-life clinical setting.

## 5. Conclusions

In multiple computer-based laboratory studies under various experimental conditions (different viewing durations, distractions, multiple patients), Visual Patient transferred more information per unit time with a reduced subjectively perceived workload and increased diagnostic certainty. Eye tracking research showed the technology works because it synthesizes and morphs vital sign information into an easy-to-interpret form. Numbers turn into colors, shapes, and rhythmic movements, a language that humans can easily perceive using the entire visual field. Qualitative research showed that users thought the technology was easy to learn and gave them a quick situation overview. The technology could be particularly useful for improving situation awareness in environments where cognitive load is high and quick and confident decisions must be made. Building on a solid theoretical foundation and our preliminary studies, we now aim to explore the value of Visual Patient technology in clinical applications.

## 6. Patents

European Union patent application 16/724016.7 “Method and system for monitoring a patient’s medical condition”; United States patent application 15/816749 “Method for monitoring and visualizing a patient’s medical condition.”; European Union trademark 1424812 “Visual Patient” Owner: University of Zurich; Swiss trademark 719318 “Visual Patient”; European Union design protection 004064178-0001 and 004064178-0002 “Visual Patient”; European patent application 19/193196 “Method and system for displaying and monitoring a patient’s blood coagulation function”; German design protection 402019100218-0001 to 402019100218-0012.

## Figures and Tables

**Figure 1 sensors-20-02112-f001:**
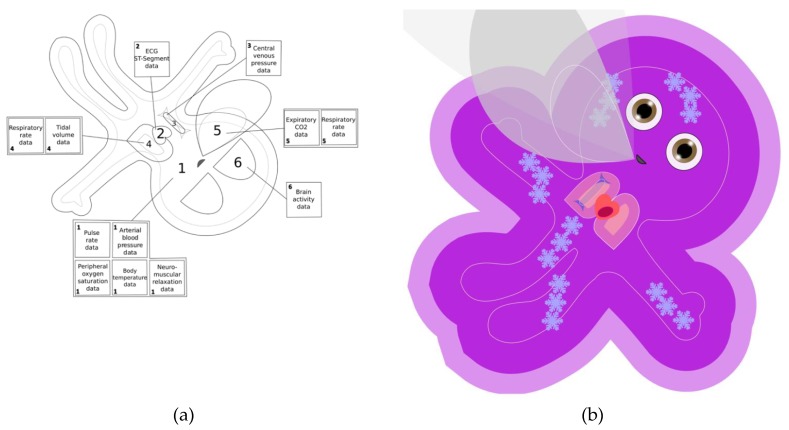
(**a**) Schematic description of the avatar. The avatar is displayed with its head toward the bottom of the monitor to represent the habitual viewpoint of an anesthesia provider during anesthesia. The numbers 1 through 6 describe the elements of the avatar: 1 = body, 2 = heart, 3 = vena cava, 4 = lungs, 5 = expiratory carbon dioxide cloud, and 6 = eyes. (**b**) A Visual Patient avatar oriented in an upward direction. This view could be used in an intensive care unit, where the practitioners usually stand at the opposite side of the patient bed.

**Table 1 sensors-20-02112-t001:** Explanations of the avatar visuals and the vital signs they represent.

Vital Sign	Visualization on the Avatar ^1^	Phenomenon in the Patient the Visualization is Modeled on
1. Pulse rate	The pulsation rate of the body (element #1) of the avatar. Example: The avatar’s body pulsates with a slow, normal, or fast frequency.	The pulsation of the arteries as visible, for example, under a microscope or palpable as a pulse.
2. Arterial blood pressure	The pulsation intensity of the body (element #1) of the avatar. Example: The avatar’s body pulsates just barely normally (i.e., reaching the white line designating the normal pulsation extension) or very intensely, extending far beyond the white “normal” line.	The strength of the pulse in the arteries depending on the arterial blood pressure as visible, for example, under a microscope or palpable as a pulse.
3. Central venous pressure	The area of the vena cava (element #3) of the avatar. Example: The size of the avatar’s vena cava is very small, normal (i.e., reaching the white line designating the normal extension), or very large, far beyond the white “normal line.”	The filling volume of the vena cava depending on central venous pressure.
4. Respiratory rate	The breathing frequency of the lungs (element #4) of the avatar and the corresponding exhalation frequency of the carbon dioxide (CO_2_) cloud (element #5). Example: The avatar’s lungs breathe with a slow, normal, or high frequency and (if data from a CO_2_ sensor is available) synchronous slow, normal, or quick exhalation of the CO_2_ cloud.	The breathing synchronous volume change of the lungs and the breathing synchronous invisible exhalation of a volume of CO_2_.
5. Tidal volume	The extension size of the lungs (element #4) of the avatar during the breathing cycle. Example: The avatar’s lungs extend just barely, normally (i.e., reaching the white line designating the normal breathing extension) or very far, noticeably beyond the white “normal” line.	The volume change of the lungs depending on tidal volume.
6. Expiratory CO_2_ concentration	The extension size of the CO_2_ cloud (element #5) of the avatar during the breathing cycle. Example: The CO_2_ cloud is just barely visible, reaches a normal extension (i.e., reaching the white line designating the normal breathing extension), or extends very wide, far beyond the white “normal” line.	The volume change of CO_2_ exhaled into the air.
7. Body temperature	The presence or absence of temperature indicators on the body (element #1) of the avatar. Example: Heat waves are rising from the avatar or ice crystals are visible on its skin.	Hyperthermia: The heat radiation from the skin. Hypothermia: The skin cold to the touch.
8. Brain activity	The form of the eyes (element #6) of the avatar. Example: The eyes of the avatar are open or closed.	High: Eyes open; pupils middle wide as in sympathetic activation. Low: Eyelids completely closed as in a sleeping patient.
9. Peripheral oxygen saturation	The color of the body (element # 1) of the avatar. Example: The avatar has a healthy skin color or a purple skin color.	Normal: Light-brown skin color according to Fitzpatrick skin type III. Hypoxia: Dark purple skin color as in cyanosis.
10. Neuromuscular function	The form of the body (element #1) of the avatar. Example: The avatar has extended extremities and an extended thumb or the extremities appear floppy.	Normal neuromuscular function: Extended extremities and thumb (healthy muscle tone in the adductor policis muscle, where care providers frequently measure relaxation). Neuromuscular block: Floppy limbs.
11. Electrocardiography ST segment	The presence or absence of a hypoxia indicator over the heart (element #2) of the avatar. Example: The heart muscle of the avatar has a homogenous red color or a dark purple spot on the heart.	Normal: Light red color of healthy myocardium. Hypoxia: Purple color of hypoxic myocardium.

^1^ The elements of the avatar are displayed in Figure 1.

**Table 2 sensors-20-02112-t002:** Summary of the reviewed literature.

Study (Year)	Study Type ^1^	Participants ^2^	Task	Method	Results
Tscholl et al. (2018) [1]	Within-subject, computer-based	Calibration and validation of avatar: 150 Comparative study: 32	Interpreting patient monitoring scenarios with Visual Patient and conventional patient monitoring	Iterative development Delphi process Rating of vital signs Rating of diagnostic certainty NASA Task Load Index	Visual Patient showed high high interrater reliability, improved vital sign perception, increased diagnostic confidence, and lowered perceived workload.
Tscholl et al. (2018) [2]	Qualitative and quantitative study	Interview part: 128 Quantitative part: 36	Providing user feedback about Visual Patient	Qualitative analysis of interviews followed by quantitative rating of statements	Visual Patient provided quick situation overview and was easy to learn
Pfarr et al. (2019) [4]	Within-subject, computer-based, eye tracking	30	Interpreting patient monitoring scenarios with Visual Patient and conventional patient monitoring with peripheral vision	Rating of vital signs Rating of diagnostic certainty Eye-tracking analysis	Visual Patient improved vital sign perception, and increased diagnostic confidence with peripheral vision
Pfarr et al. (2019) [6]	Within-subject, computer-based	38	Interpreting patient monitoring scenarios with Visual Patient and conventional patient monitoring under distraction	Rating of vital signs NASA Task Load Index	Visual Patient improved vital sign perception and reduced workload under distraction
Garot et al. (2020) [5]	Within-subject, computer-based	38	Interpreting multiple-patient monitoring scenarios with Visual Patient and conventional patient monitoring	Rating of vital signs NASA Task Load Index	Visual Patient improved vital sign perception and reduced workload under distraction except in 30 s scenarios
Tscholl et al. (2020) [3]	Within-subject, computer-based, eye-tracking	30	Interpreting patient monitoring scenarios with Visual Patient and conventional patient monitoring	Eye-tracking analysis	Visual Patient enabled parallel perception of vital signs as a result of its visual design
Rössler et al. (2020) [7]	Between-subject, computer-based	42	Interpreting patient monitoring scenarios with Visual Patient and conventional patient monitoring	Rating of vital signs	Class-based and individual instruction both feasible for Visual Patient training

^1^ All studies were two-center studies, except Rössler et al. [7], which was a single-center study. ^2^ Participants were anesthesiologists and certified nurse anesthesiologists for all studies except Rössler et al. [7], in which all participants were certified nurse anesthetists.

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
