# Peer review of "Situation Awareness-Oriented Patient Monitoring with Visual Patient Technology: A Qualitative Review of the Primary Research"

_sensors, 2020, doi:10.3390/s20072112_

Round 1

Reviewer 1 Report

the manuscript approaches a relevant aspect of patient management looking for innovative technical support.

in the attached file the point by point comments are reported.

Author Response

Comment:

the authors approached a relevant aspect of patient management in surgery and intensive care unit. Visual Patient Technology may enormously support the clinical activity avoiding work overload and misunderstanding.

Response:

Thank you very much for your work on our manuscript and your kind words.

Comment:

however, the oversimplification of the information provided may affect the therapeutic decision. in medical practice it is frequent the condition where the clinical status gradually change without a clear yes/not datum. accordingly, more comparative data should be provided in order to test the real usefulness in clinical practice

Response:

Thank you very much for this comment. You address a crucial aspect, which we are considering to a greater extent in the discussion in this revision. Situation awareness has three levels—Recognition, understanding, and foresight. The studies summarized in this review showed that Visual Patient in its current form mainly improves situation awareness levels 1 and 2 (the technology makes more information perceptible per time). The information flowing into the version of Visual Patient tested in the reviewed studies was a snapshot of the status quo, the current patient situation. However, this is about to change as we are currently conducting a study evaluating a trend forecast for the Visual Patient. At the same time, we are conducting a simulator study with anesthesia teams in Zurich to evaluate the benefits of Visual Patient technology in a realistic clinical situation. Our goal is to publish these papers no later than 2021, and this review article will be the foundation upon which these two papers will build. We now highlight the limitations of Visual Patient in a separate, new section of the paper (Section 2.6 - Limitations of Visual Patient).

Reviewer 2 Report

This is a very well-structured and organized paper, and I only have one request to make. Possible drawbacks of the visual patient technology are not described to the extent I would have liked to see. For questions/concerns like the following ones an answer should be given, either by making a solid argument why the concern is insubstantial, by referencing pertinent results in the literature, or by pointing to planned studies.

  1. In analogy to the ideal monitor's goal "one that does not create false alarms": How sensitive is Visual Patient to wrong measurements and artifacts? Typically those present themselves as sharp jumps of numerical measurements and one would assume that the visual representation of these jumps may have an irritating effect on the user.
  2. For some physiological parameters, for instance the respiratory rate, it might take a longer time to get the correct information than by reading the number from a monitor. In a case of a low RR (e.g. 6/min) it will take 10-20 seconds to get the right impression, and the low RR might even be missed by a short look at the display.
  3. An important drawback of the classical monitor interface is that recognition of clinically meaningful patterns of parameter changes is not enforced. Is there any evidence that such patterns (e.g. high HR and low BP, but also patterns shown in, say, Lynn 2011 are easier detected than by using classical monitor UIs?
  4. As mentioned in the paper, classical monitors (to a degree) utilize trend display of parameters. Such trends also support a projection of a patient's likely trajectory by extrapolation. The reader would be interested in understanding whether the lack of a comparable functionality in Visual Patient (which is the reviewer's assumption to be present) is not clinically important
  5. Classical Monitors not only show continuous parameters, but also events, like asystole, apnea and many other arrhythmia events. IS the Visual Patient technology of incorporation such events, and if not: Why is this not deemed to be a problem?

Minor points:

  • "An exception to this is the display of trend information, which shows the course of the measured values as graphic bands." (1.2.2.). It should be mentioned that there are more advanced visualizations which aggregate several measurements into one picture, e.g. the monitors in the authors' institution have a graphical map for the ST-Values (https://www.philips.ch/healthcare/product/HCNOCTN176/stmap) which supports a quick visual assessment of ST-segment changes and there localizations.
  • The problem of alarm fatigue in the OR is likely to be mitigated by Visual Patient. I would have loved to see this likely effect to be at least mentioned. Furthermore, the references to literature on alarm fatigue are limited to distractions in anesthesie which doesn't cover the full breadth of the problem. Below are some suggestion for papers which might be considered.

  • LYNN, Lawrence A.; CURRY, J. Paul. Patterns of unexpected in-hospital deaths: a root cause analysis. Patient safety in surgery, 2011, 5. Jg., Nr. 1, S. 3.  
  • Alarm Fatigue in the OR - Pertinent literature suggestions: Schmid, Felix, et al. "The wolf is crying in the operating room: patient monitor and anesthesia workstation alarming patterns during cardiac surgery." Anesthesia & Analgesia 112.1 (2011): 78-83.
  • Beatty, P. C. W., and S. F. Beatty. "Anaesthetists' intentions to violate safety guidelines." Anaesthesia 59.6 (2004): 528-540.
  • Weaver, Joel M. "Alarm fatigue can decrease the safety of dental office sedation and anesthesia." Anesthesia progress 60.3 (2013): 93.

Author Response

Comment:

This is a very well-structured and organized paper, and I only have one request to make.

Response:

Thank you very much for your work on our paper and your sound and helpful suggestions.

Comment:

Possible drawbacks of the visual patient technology are not described to the extent I would have liked to see. For questions/concerns like the following ones an answer should be given, either by making a solid argument why the concern is insubstantial, by referencing pertinent results in the literature, or by pointing to planned studies.

  1. In analogy to the ideal monitor's goal "one that does not create false alarms": How sensitive is Visual Patient to wrong measurements and artifacts? Typically those present themselves as sharp jumps of numerical measurements and one would assume that the visual representation of these jumps may have an irritating effect on the user.

Response:

Thank you very much for this comment. We incorporated your feedback into the discussion and considered potential drawbacks in more detail in this review. It is true that artifacts are a constant problem in monitoring and are recognized as such by anesthetists and operating theatre staff. In one of our studies, 64% of the respondents agreed with the statement: "Artifacts complicate the understanding of the situation." In the Visual Patient version used in the studies we summarized in this review, we configured Visual Patient with similar alarms as the acoustic ones. Whenever an audible alarm went off in the conventional examples, Visual Patient also showed the appropriate alarm state. In the future, Visual Patient may be configured more conservatively or more liberally than the acoustic thresholds in the monitor.

Moreover, the technology could serve as a platform for a machine learning algorithm. We are currently evaluating an algorithm for Visual Patient, which, based on the trend in the last few measurements of a vital sign, already indicates the state that is likely to occur in the very near future. This excursion serves to illustrate the range of possibilities of the front-end of the technology. Now to the question of visual alarms, which might disturb users. Whatever choice of Visual Patient configuration used in the product in a hospital, the technology should remain configurable by the user. We think this is an essential requirement for it to reach its goal of becoming a successful tool for situation awareness. We consider it a positive aspect if a visual alarm attracts users' attention. The logical consequence then is to either adjust the alarm limit, for example, because the patient's vital signs are different from the preset normal range or do something to correct the situation.

Artifacts do happen, and it would be nice if technology would get to a point where it could filter them out. But compared to many real alarms, there are relatively few artifact alarms. In our opinion, it is crucial to be able to display the right alerts clearly and, at the same time, to invent algorithms and other technologies that reduce artifacts.

On a side note, from our first personal experience with the technology, we can say that the Visual Patient's visual cues are much less disturbing than the acoustic bells because when looking at Visual Patient, it is immediately apparent which alarm is going off. When looking at a monitor that displays a lot of numbers, sounds and blinks, situation awareness is much less clear and, therefore more stressful.

We discuss Visual Patient in connection with Alarm Fatigue in the newly added section 4.5.2 in the discussion of this revision.

Comment:

2. For some physiological parameters, for instance the respiratory rate, it might take a longer time to get the correct information than by reading the number from a monitor. In a case of a low RR (e.g. 6/min) it will take 10-20 seconds to get the right impression, and the low RR might even be missed by a short look at the display.

Response:

That's an excellent comment. We came across precisely this problem during the initial testing of our technology. We solved it by adding shadows that show the maximum of a previous extension and remain visible until the peak of the next expansion. Specifically, this problem occurs with respiratory rate and blood pressure. We implemented a shadow, which designates the maximum of the current extension of an element. We added this modification for all vital signs, which are visualized through the extension of a component to enable instant recognition, at any point during the pulsation- or breathing cycle. We based this modification on the feedback of participants indicating that a slow pulse increases the time required to recognize blood pressure, and a slow respiratory frequency increases the time necessary to understand tidal volume and expiratory CO2. We also explain this feature in Supplementary Video 1 and now included it in the review of our first study in this revision.

Comment:

3. An important drawback of the classical monitor interface is that recognition of clinically meaningful patterns of parameter changes is not enforced. Is there any evidence that such patterns (e.g. high HR and low BP, but also patterns shown in, say, Lynn 2011 are easier detected than by using classical monitor UIs?

Response:

Thank you very much for this excellent comment. A working tool for pattern recognition, as shown in the Lynn et al. paper, can offer a variety of possibilities for earlier identification of patterns in patient monitoring. We have not yet systematically investigated the value of Visual Patient regarding pattern recognition. Still, we noticed some things that worked very well with our participants due to the anatomical shape of Visual Patient. As two examples, we would like to mention the following. A too fast ECG heart rate, which the current version indicates as a fast pulsation of the heart together with a regularly and normal frequency pulsating Visual Patient body, suggests a heart rhythm with an irregular neuromuscular coupling of the myocardium. A patient with a relaxed body but open eyes indicates the occurrence of intraoperative awareness. The Platform Visual Patient, therefore, offers a variety of possibilities for improved pattern recognition. We discuss Visual Patient in connection with pattern recognition in the newly added section 4.5.4 in the discussion of this revision.

Comment:

4. As mentioned in the paper, classical monitors (to a degree) utilize trend display of parameters. Such trends also support a projection of a patient's likely trajectory by extrapolation. The reader would be interested in understanding whether the lack of a comparable functionality in Visual Patient (which is the reviewer's assumption to be present) is not clinically important

Response:

The authors thank you very much for this comment. We have several points of view in this regard, which we will now discuss further in the discussion. One aspect is that Visual Patient in its current form should by no means completely replace regular monitoring, but only supplement it in those areas where routine monitoring has deficits (quick and easy situation overview). The trends of routine patient monitoring are significant for situation awareness and cannot be replaced by Visual Patient in its current version. The second, more exciting view is that we will begin a study this summer about a trend function for Visual Patient named Visual Patient Trend Preview. This technology uses a simple algorithm to calculate from a vital signs recent past (a few seconds to a few minutes, depending on the vital signs) the length of time until this vital sign will reach the defined threshold value due to the current change.  The shorter this time is, the more prominent Visual Patient will integrate this next status of the vital sign into the current one. Imagine a Visual Patient that pulsates a few beats more intensively than usual every few seconds. This would indicate that the direction of the blood pressure is going up. This technology could, in theory, enable Visual Patient only patient monitoring. Please, expect a publication regarding this development in 2021 at the latest. This review will serve as the basis for these upcoming developments.

Based on this comment, we now present and discuss Visual Patient in connection with trend monitoring in the newly added section 4.5.3 in the discussion of this revision.

Comment:

5.  Classical Monitors not only show continuous parameters, but also events, like asystole, apnea and many other arrhythmia events. Is the Visual Patient technology of incorporation such events, and if not: Why is this not deemed to be a problem?

Response:

This comment also is excellent. Thank you very much. Of course, we want and will include this information in Visual Patient. Asystole and apnea already appear in Visual Patient. Visual Patient then does not pulsate or breathe. A part of the planned study described in the section above will test intensive care extensions for Visual Patient. These extensions include an upright and enlarged Visual Thorax, which shows the anatomically correct display of  ST-segment deviations and different displays for electrical heart activity (and pacemaker) in addition to the PICCO parameters. Asystole and apnea can already be displayed in Visual Patient today. Visual Patient then does not pulsate or does not breathe. Users can discern this situation from the case of a disconnected ECG because the heart and lungs are not greyed out and dotted but vivid. That means cables are connected, the heart is not beating/the patient is not breathing. In the future possibilities for the Visual Patient platform seem endless, just think of the information from the electronic patient data management system integrated into the visualization. We will finally be able to see at a glance where a patient's installations are. We now also address this issue in more detail in the discussion of this revision. Based on this comment, we now present and discuss Visual Patient in connection with event monitoring in the newly added section 4.5.5 in the discussion of this revision.

Comment:

Minor points:

  • "An exception to this is the display of trend information, which shows the course of the measured values as graphic bands." (1.2.2.). It should be mentioned that there are more advanced visualizations which aggregate several measurements into one picture, e.g. the monitors in the authors' institution have a graphical map for the ST-Values (https://www.philips.ch/healthcare/product/HCNOCTN176/stmap) which supports a quick visual assessment of ST-segment changes and there localizations.

Response:

Many thanks for this valuable comment. We have included the ST-Map in section 4.8 of this revision.

Comment:

  • The problem of alarm fatigue in the OR is likely to be mitigated by Visual Patient. I would have loved to see this likely effect to be at least mentioned. Furthermore, the references to literature on alarm fatigue are limited to distractions in anesthesie which doesn't cover the full breadth of the problem. Below are some suggestion for papers which might be considered.
  • LYNN, Lawrence A.; CURRY, J. Paul. Patterns of unexpected in-hospital deaths: a root cause analysis. Patient safety in surgery, 2011, 5. Jg., Nr. 1, S. 3.  
  • Alarm Fatigue in the OR - Pertinent literature suggestions: Schmid, Felix, et al. "The wolf is crying in the operating room: patient monitor and anesthesia workstation alarming patterns during cardiac surgery." Anesthesia & Analgesia 112.1 (2011): 78-83.
  • Beatty, P. C. W., and S. F. Beatty. "Anaesthetists' intentions to violate safety guidelines." Anaesthesia 59.6 (2004): 528-540.
  • Weaver, Joel M. "Alarm fatigue can decrease the safety of dental office sedation and anesthesia." Anesthesia progress 60.3 (2013): 93.

Response:

Thank you very much for this comment and the literature suggestions, some of which we included in the revised manuscript. We now also present and discuss Visual Patient in connection with alarm fatigue in the newly added section 4.5.2 in the discussion of this revision.

Reviewer 3 Report

This paper provides an overview of an interesting research area: Visual Patient Technology. The paper is well written and analyzes deeply the main concepts and findings of the state of the art. I liked the area (the idea behind the proposed technology) and the paper itself. However, I have some comments about the content of the paper.

Suggestions and questions:
1- The paper does not present its contributions. A paragraph (in the introduction section) to explicitly show the contributions of the paper should be provided. This comment is also to allow the authors to provide content that shows the relation between "Visual Patient Technology" and "Sensor area", which is not explicitly presented in the paper.
2- Add a paragraph at the end of the introduction section to show the paper organization.
3- Legend of Figure 1 is too long. That text would be better as a paragraph describing the figure. Legend could be more direct.
4- I think the names of the journals where papers were published could be omitted - section 3. This information can be consulted in the references. So, only the reference is enough.
5- The authors propose a paper that provides an overview of their published works. How can the authors claim that there is no more related research published? Is section "4.7. Overview of Other Studies" enough? I mean, why did the authors not perform a systematic literature review? It would be interesting to confirm the absence of other very related materials.
6- What are the gaps in the research area? What are the limitations in the "Visual Patient Technology" that researchers from the "sensor area" should investigate and propose solutions? I mean, what are the concerns? What should be done in the future? Section "4.6. Strengths and Limitations of the Reviewed Studies" is poor and fails to present such content.

Author Response

Comment:

This paper provides an overview of an interesting research area: Visual Patient Technology. The paper is well written and analyzes deeply the main concepts and findings of the state of the art. I liked the area (the idea behind the proposed technology) and the paper itself. However, I have some comments about the content of the paper.

Response:

Thank you very much for your kind words and your work with our manuscript.

Comment:

Suggestions and questions:

1- The paper does not present its contributions. A paragraph (in the introduction section) to explicitly show the contributions of the paper should be provided. This comment is also to allow the authors to provide content that shows the relation between "Visual Patient Technology" and "Sensor area", which is not explicitly presented in the paper.

Response:

Thank you very much for this comment. We state the paper's contributions in the introduction and also put the paper into context with sensors.

Comment:

2- Add a paragraph at the end of the introduction section to show the paper organization.

Response:

Thank you very much for this comment. We have now added the proposed description of the organization of the paper in the introduction.

Comment:

3- Legend of Figure 1 is too long. That text would be better as a paragraph describing the figure. Legend could be more direct.

Response:

Thank you for this advice. We have now shortened the legend considerably and transferred most of it to the text.

Comment:

4- I think the names of the journals where papers were published could be omitted - section 3. This information can be consulted in the references. So, only the reference is enough.

Response:

We have omitted the journal names in this revision in accordance with this proposal.

Comment:

5- The authors propose a paper that provides an overview of their published works. How can the authors claim that there is no more related research published? Is section "4.7. Overview of Other Studies" enough? I mean, why did the authors not perform a systematic literature review? It would be interesting to confirm the absence of other very related materials.

Response:

Thank you very much for this comment. We have been familiar with the literature, and also intellectual property publications about this subject for several years. Furthermore, we regularly scan the literature looking for similar developments. While there are a lot of papers on novel display technologies, directly comparable papers, which used an avatar or an anatomic model to display vital sign information are rare. In 2006, Drews et al. (https://journals.sagepub.com/doi/abs/10.1518/001872006776412270) published a review article on this topic, and since then, progress in the field was relatively slow. We are confident we listed the most recent ones in our overview. We have adapted the corresponding parts in the discussion section of the manuscript.

Comment:

6- What are the gaps in the research area? What are the limitations in the "Visual Patient Technology" that researchers from the "sensor area" should investigate and propose solutions? I mean, what are the concerns? What should be done in the future? Section "4.6. Strengths and Limitations of the Reviewed Studies" is poor and fails to present such content. 

Response:

Thank you very much for this comment. We have now made the points discussed clearer and more comprehensive. We also created a separate section with the limitations of the current Visual Patient version. We hope this is an improvement over our first version.

Round 2

Reviewer 1 Report

The Authors provided with accurate revision of the manuscript, improving it.

Reviewer 3 Report

Thank you for addressing the comments and making the requested changes.

All of my concerns were resolved. So, the paper is suitable to be published.